# Investing in the Early Childhood Mental Health Workforce Development: Enhancing Professionals’ Competencies to Support Emotion and Behavior Regulation in Young Children

**DOI:** 10.3390/brainsci7090120

**Published:** 2017-09-19

**Authors:** Shulamit N. Ritblatt, Audrey Hokoda, Charles Van Liew

**Affiliations:** 1Child and Family Development, San Diego State University, San Diego, CA 92182-4502, USA; audrey.hokoda@mail.sdsu.edu; 2Humanities Department at Grand Canyon University, Phoenix, AZ 85017, USA; cavanliew@yahoo.com

**Keywords:** early childhood, mental health, challenging behaviors, professional training, social emotional development, prevention, early intervention, workforce development

## Abstract

This paper delineates a preventive approach to early childhood mental health by preparing the workforce to provide relational, sensitive care to young children ages 0–5. One of the most prevalent issues in early childhood is behavioral challenges and the inability of young children to regulate themselves. This leads to an expulsion rate in early childhood (3–4 times higher than K-12 expulsion rate) and future mental health issues. The Early Childhood Social-Emotional and Behavior Regulation Intervention Specialist (EC-SEBRIS) graduate level certificate program was created to strengthen early care and education providers with the knowledge and practice of how to support emotion and behavior regulation in young children in their groups. Evaluation data provide evidence that early care and education professionals increased in their perception of self-efficacy and in their sensitivity of care and skills to support behavioral health in young children. Results indicated that the children in their care showed less challenging behaviors and increased social competencies. This manuscript highlights the importance of prevention and the dire need to provide young children with high-quality, appropriate care to support their mental health.

## 1. Introduction

There are nearly 11 million children under the age of five in the USA whose parents are working while other adults care for them. These children spend an average of 36 h a week in child care and 25% of the children, nearly 3 million, have multiple childcare arrangements as their parents work longer hours and have non-traditional work schedules [1]. As children spend more and more hours in childcare, it becomes increasingly important that early childhood professionals are well educated and trained in recognizing and responding to early signs of social-emotional and mental health problems [2].

Prevalent, serious problems in early childhood include behavioral challenges and the inability to regulate emotions. These behavioral challenges (e.g., aggression, poor self-control, noncompliance, and difficulties socializing with peers) occur in the classrooms of young children once every six minutes [3], and lead to an expulsion rate in preschool settings three to four times higher than K-12 expulsion rates [4,5]. Difficulties regulating emotions and behaviors are associated with internalizing and externalizing behavioral problems in young children that may result in lifelong psychiatric disorders and maladjustments [6,7,8]. For example, poor emotion regulation is associated with depression, anxiety, schizophrenia, and post-traumatic stress disorder [9,10,11], sexual promiscuity [12], and physical health issues [13,14,15]. If left untreated, young children with challenging behavior are more likely to display at-risk behaviors as adolescents, such as delinquency, gang involvement, and substance abuse; in fact, early childhood behavior problems are the single best predictor for several of these long-term outcomes [16].

Emotional and behavioral regulation in young children is critical to healthy mental development and research suggests that this regulatory skill is acquired via emotionally attuned adult–child interactions. In the absence of these interactions, the ability to regulate emotion is impaired as well as the architecture of the developing brain [17,18]. Early positive experiences support the creation of circuits in the brain that are responsible for generating emotions, behavioral responses, perception, and bodily sensations [19]. Studies have documented the impact of early childhood stress (e.g., adverse childhood experiences) on neurobiological development that has long-term effects on anxiety, aggression, self-regulation, cognition, and social-emotional functioning (e.g., [20,21]).

Growing evidence suggests that the prevention and early intervention are needed particularly for young children who experience prolonged toxic stress. Research also demonstrates that, when early childhood stress is experienced, stable supportive relationships with adults can buffer these harmful effects and facilitate adaptive coping responses. “To that end, there is a critical need for creative new interventions that strengthen the capacity of parents and other caregivers to reduce sources of excessive adversity and to help build effective coping skills in children who experience high levels of stress” [22] (p. 1645).

The purpose of this paper is (a) to highlight research emphasizing the critical role of adult–child relationships as the foundation of mental health and a buffer to negative experiences and toxic stress; (b) to describe a training model (Early Childhood Social Emotional and Behavior Regulation Intervention Specialist, EC-SEBRIS, program), a preventive approach to early childhood mental health that prepares the workforce to provide relational, sensitive care to young children ages 0–5; and (c) to provide outcome data indicating the effects of the training on the early childhood professionals participating in the year-long program and the children they served.

### 1.1. The Importance of Adult–Child Relationships and Early Childhood Care and Education in Promoting Mental Health

An abundance of translational, cross-disciplinary studies (e.g., [23]) have helped us understand young children’s development and mental health are outcomes of an ongoing interaction between nature and nurturer [24,25]. This biodevelopmental approach [26] recognizes the link between emotion, the brain, and the body, and highlights the importance of positive interactions and stable relationships in meeting the specific developmental needs of the young child [27,28,29,30,31].

Poor quality of care in early years is a major contributor to toxic stress [32]. Continuous elevated levels of the stress hormone, cortisol, have permanent negative changes on the brain [33]. Research findings [34,35] provide support for the notion that sensitive and responsive caregiving helps children express distress without elevating cortisol levels while anticipating adults’ help.

A critical piece of the puzzle in providing high quality relationships lies with the early childhood teachers with whom many children spend a majority of their time. It takes one caring adult who has a positive relationship with a child to protect the child, especially those children who experience multiple risks and stressors [36]. These positive out-of-home relationships can change the attachment internal working models developed by the children with their parents and buffer them from the risks of adverse caregiving received in the home [37].

Multiple studies have been conducted on teacher–child relationships and the results point to the overwhelmingly positive and lasting effects of close teacher–child relationships on children emotionally, socially, and academically [38,39,40,41,42,43,44,45]. For example, positive sensitive teacher–child relationships appear to decrease behavioral challenges and the negative effects for children at risk for externalizing and internalizing problems [38,46]. Similarly, numerous studies on positive behavior support in early childhood education settings indicate that effective behavior management in the classrooms support young children’s abilities to regulate their emotions and behaviors and yield better child outcomes [47,48,49,50].

It is important to understand that the stress, burnout, and frustration early childhood educators feel in their jobs affect their relationships with the children in their care and the environments they create for the children. Educators who experience a high level of stress are less likely to engage in warm, responsive caregiving and experience more negative behaviors in the classroom [51,52]. Early childhood professionals point out that dealing with challenging behaviors is their number one stressor, and teachers also report feeling unprepared to support emotion and behavior regulation and that they experience a high level of frustration and helplessness on a daily basis.

An effective way to enhance early childhood professionals’ capacity to provide warm, responsive caregiving and reduce problem behaviors and rates of expulsion is giving professionals greater access to mental health consultation [5,53,54,55]. Unfortunately, only 23% of teachers in an early childhood settings report regular access to a mental health consultant [56]. Furthermore, more frequent consultations are associated with lower turnover of teachers, improved effectiveness of teachers, and an enhanced program quality [53,57,58]. Research further indicates that, when mental health consultants are housed within the early childhood education programs, they are found to be more effective in their work with children and teachers [59,60,61,62]. In addition to mental health consultation, it is recommended that reflective practice is included in educational programs and trainings for caregivers and early childhood educators [63]. Insightfulness is critical to the understanding of the child’s internal world, enhances caregivers’ sensitivity, and facilitates the development of secure attachments [64,65,66,67].

The research presented above shows that early experiences have long-lasting effects on the developmental trajectory of the child and their mental health, and that nurturing, sensitive, and responsive relationships between adults and young children are critical in ensuring the provision of high-quality care [22]. Prevention and early intervention is needed particularly for young children who experience adversity and toxic stress [22], and quality early childhood care and education can serve as a buffer and promote healthy mental development in young children. Specifically, professional training that focuses on emotional and behavior regulation support is essential for early childhood professionals working with young children [68]. The following section describes an innovative, cutting-edge training program designed to prepare a workforce of early childhood professionals to provide sensitive relational care and to address early mental health problems with behavior and emotional regulation.

### 1.2. Training Model: Early Childhood Social Emotional and Behavior Regulation Intervention Specialist (EC-SEBRIS) Program

The underlying assumption that has guided the development and implementation of the Early Childhood Social-Emotional and Behavior Regulation Intervention Specialist (EC-SEBRIS) Certificate Program ([69] is that the early childhood professional is an important member of the first-response “mental health” team to manage children’s needs and daily care. The EC-SEBRIS was designed to help teachers and early childhood professionals develop the skills and competencies needed to address challenging behaviors in their classrooms or at homes, so that they can meet the critical social-emotional and behavioral needs of young children.

Therefore, the purpose of the EC-SEBRIS Certificate Program is to establish a recognition and response model to meet the needs of increasing numbers of young children who attend childcare programs and have social-emotional and behavioral challenges and to help early childhood professionals be more self-aware and emotionally available to provide the children under their care with sensitive, warm, and responsive positive interactions. The program is based on the Teaching Pyramid (National Center in the Social and Emotional Foundations for Early Learning—CSEFEL), and has three levels of training teachers to address emotional and behavioral problems among young children. The first level—*Promotion (Tier I)—*a prevention approach that targets children who are at risk of poor developmental outcomes, including early identification using screenings and improving the quality of child care (ages 0–5) that encourages brain development, learning, and emotional well-being of all young children. The second level—*Preventive Intervention (Tier 2)—*targets children who already exhibit emotional and behavioral problems and provides them and their families with highly specialized levels of service that mitigate the effects of risk and stress and address potential early relationship challenges or vulnerabilities that affect early development. The third level—*Treatment (Tier 3)—*targets children in distress or with clear symptoms indicating mental health disorder to provide them and their families with highly specialized levels of service by skilled staff [68].

The EC-SEBRIS Certificate Program targets early childhood professionals (i.e., students) in the field who have earned an undergraduate degree in Child Development or related areas. Following the recommendations described in the Delivery of Infant-Family and Early Mental Health Services Training Guidelines and Personnel Competencies proposed by California’s Infant, Preschool and Family Mental Health Initiative (2010), it uses a wraparound triple method of teaching that includes three main domains: Knowledge, Experience, and Reflective Supervision (see Figure 1).

As depicted in Figure 1, the *Knowledge* domain is covered in 4 seminar courses (over the course of two consecutive semesters): (1) Seminar in Child Development Theories-Intervention and Prevention, which reviews attachment and affect regulation theories, assessment, and screening information for differentiating typical and atypical development in young children (0–5) and strategies for implementing a regulation plan at childcare sites; (2) Seminar in Human Development: Positive Behavior Support for Young Children with Challenging Behavior, which teaches students how to implement best practice interventions that promote social-emotional and behavioral development in young children (0–5); (3) Eco-behavioral Assessment and Intervention, which reviews eco-behavioral assessment techniques (e.g., direct observations, rating scales, structured interviews, analogue measures) and behavior analytic intervention strategies promoting social-behavioral development; and (4) Advanced Applied Behavior Analysis, which reviews Applied Behavior Analysis (ABA) with a focus on teaching positive social interaction and self-management skills to young children.

As part of the *Experience* Domain, students were required to work a minimum of 20 h a week providing direct services/education to children 0–5 years of age. The majority of the students enrolled in the program had a job in which they provided direct service/education to children of this age group. These work sites were considered as the students’ practicum sites where they could apply and practice the knowledge and skills learned in their courses. The underlying notion was that the application of knowledge at the same site that the student spent most of their time working with young children can benefit the children and can have a ripple effect on the other providers at the site. The sites included early childhood education programs, agencies providing mental health and behavior support services, home visitation programs, and others.

The *Reflective Supervision* domain includes the following components: reflective facilitation for students (group and individual), video feedback, and on-site coaching. This triple method of coaching and mentoring supports the reflective process and provides the student with an opportunity to identify their own needs and potential areas of growth [70,71,72]. According to The California Infant, Preschool and Family Mental Health Initiative: Training Guidelines and Recommended Personnel Competencies [73], reflective supervision is essential when working with young children. Research suggested that it can enhance caregivers’ sensitivity. Hence, experienced licensed clinical supervisors provided bi-monthly individual supervision and weekly group supervision. During group supervision, students practice concepts and theoretical information presented in the foundation courses. Students “learn by doing” and by discussing their experiences using their video recordings with other students [63].

As mentioned above, students in the EC-SEBRIS Certificate Program were equipped with a video camera and were asked to videotape their interactions with children once a week for 30–45 min. Videotaping of teacher–child interactions has been found to support the reflective process needed for the adult to be more aware and sensitive when interacting with young children [74]. Viewing these recordings with their peers and the clinical reflective supervisor provided students the opportunity to “see” themselves as they interact with children and be aware of the transactional nature of their interactions with the children.

The Reflective Supervision domain included on-site coaching [75,76,77]. The coaches were highly experienced early childhood intervention professionals who have had extensive experience working with children, families and other professionals in the early childhood intervention and education field. The visits of onsite coaches took place four to five times a year (two to three times each semester). These coaches assisted students in assessing their initial skills level and establishing goals and objectives for future professional growth and self-improvement.

### 1.3. Evaluation of EC-SEBRIS Program

A third goal of this paper is to present outcome data evaluating the effects of the training on the early childhood professionals who graduated from the program and the children they served. Specifically, this study examines the following questions: (1) Did the students enrolled in EC-SEBRIS Certificate Program show an increase in their ability to provide sensitive, responsive care and support behavioral and emotional regulation in young children? (2) Did children improve in their social-emotional functioning after the intervention was offered to them by the students as reported by parents?

## 2. Materials and Methods

This study evaluates the effectiveness of the Early Childhood Social-Emotional and Behavior Regulation Intervention Specialist (EC-SEBRIS) Certificate Program. Questions focus on whether the students enrolled in the program show an increase in their ability to offer sensitive, responsive care, provide behavior support, and promote social emotional development, and whether the children improved in behavior and emotion regulation after the students worked with them. The study used mixed methods (observations, surveys) and multiple sources (students, site supervisors, parents).

### 2.1. Participants

#### 2.1.1. Students

The sample consisted of 154 individuals (93.51% female; 6.49% male). The age group distribution of the sample was 46.10% were between 18 and 24 years, 22.73% were between 25 and 29 years, 15.58% were between 30 and 39 years, 8.44% were between 40 and 49 years, 5.19% were between 50 and 59 years, and 1.95% were 60 years or older. Ethnically, 45.45% identified as White, 35.71% identified as Hispanic, 8.44% identified as Black, 4.55% identified as Asian or Pacific Islander, 2.6% identified as Middle Eastern, and 3.25% identified as multi-ethnic. Ninety-five students were monolingual and 59 students were at least bilingual. English was the most common primary language (72.08%), followed by Spanish (21.43%) and other (6.49%).

The evaluation of the EC-SEBRIS certificate program started with the 2nd Cohort and proceeded into a more comprehensive evaluation with the third Cohort, hence the evaluation focused on 4–5 cohorts. 22 to 35 students graduated each year from the certificate program. Demographics of the cohorts under evaluation (see Table 1) indicate that there were no significant differences among these cohorts in these domains per analyses.

#### 2.1.2. Supervisors

Sixty-four (64) supervisors of students enrolled in the certificate program were asked to complete surveys assessing the competencies of students and how the student’s enrollment in the EC-SEBRIS Certificate Program affected their practicum site. The students worked as master teachers in preschools, behavioral health specialists, home visitors, early interventionists, special educators, and directors of agencies. The supervisors were asked to complete a survey at the beginning of the first semester (September/October) and at the end of the second semester (April/May).

#### 2.1.3. Parents

Parents of children at students’ practicum sites whose child received behavior regulation support from the student were asked to rate their children’s social competencies and behavioral challenges prior to the start of the intervention (pre- at the beginning of the first semester, October–November) and then respond again to the survey at the end of the second semester (post-April–May). Sixty-three (63) parents completed a pre-post survey about the emotional and social skills of their child. The lower number of parent respondents is due to the transient nature of children placed in early childhood setting or coming to services. Often, a student would start working with a specific child in the fall and had to switch and work with another child in the spring. This means that the pre-survey done in the fall did not match the post-survey done in the spring by another parent of a different child. Hence, the total number of matching pre-post is 63.

### 2.2. Measures

#### 2.2.1. The Caregiver Interaction Scale (CIS) 

Quality of care provided by early childhood professionals was measured using The Arnett Caregiver Interaction Scale (CIS) [78]. This tool measures the verbal and physical interactions between the providers and the children in their classrooms. It is designed for use during observation of teacher caregiving skills. The CIS contains 26 items, categorized into four subscales: teacher sensitivity (10 items), harshness (9 items), detachment (4 items), and permissiveness (3 items) towards children [79]. Items on this scale are rated on a 4-point scale of *Not at All, Somewhat, Quite a Bit, and Most of the Time* [79]. These items are grouped into the four subscales and mean scores are calculated to determine caregiver interactions [80].

To assess the level of sensitivity and positive interactions which are critical to mental health, observations were performed to determine whether there were changes in sensitivity amongst students’ overtime based on scores collected from the CIS, specifically in the area of measure categorized as “sensitivity”. The sensitivity section of the CIS examines whether or not the student, “speaks warmly to the children”, “listens attentively when children speak to her”, “seems to enjoy the children”, “when the children misbehave, explains the reason for the rule they are breaking”, “encourages the children to try new experiences”, “seems enthusiastic about the children's activities and efforts”, “pays positive attention to the children as individuals”, “talks to children on a level they can understand”, “encourages children to exhibit prosocial behavior, e.g., sharing, helping”, and “when talking to children, kneels, bends, or sits at their level to establish better eye contact”. The Arnett Caregiver Interaction Scale was internally consistent, with Cronbach’s α = 0.85 and 0.86 at baseline and follow-up, respectively.

#### 2.2.2. Supervisor Survey

This 10-item Likert-scale survey ask supervisors how the student’s enrollment in the EC-SEBRIS Certificate Program has impacted the practicum site (e.g., childcare center, behavioral services, or kindergarten classroom). The pre-post scale provides an opportunity for the supervisor to rank the student on a scale from 1 to 4 (1 = Strongly disagree; 2 = Disagree; 3 = Agree; 4 = Strongly agree) on the areas of knowledge and experience that the students in the program are believed to be focusing on in their classes and reflective supervision. The supervisor survey was developed partially following the Goal Achievement Scale (GAS)-ECE Center Director [57]. The GAS tool was administered to child care directors and teachers to measure teachers’ competencies on general mental health activities or program goals. The supervisor survey includes 10 items such as “Student understands children’s social and emotional development”; “Student tries to understand the meaning of children’s behavior”; “Student is able to manage children’s challenging behaviors”; “Student responds appropriately and effectively to children in distress”; “Student is capable of reflecting on experiences, thoughts and feelings involved in working with infants, young children and families”. The supervisor survey also exhibited good internal reliability, with Cronbach’s α = 0.91 and 0.91 at baseline and follow-up, respectively.

#### 2.2.3. Self-Efficacy (SE) Survey

A self-efficacy measure assessing students’ confidence in achieving the evidence-based competencies for promoting social and emotional development and addressing challenging behavior in young children (described by the National Center on the Social and Emotional Foundations for Early Learning; CSEFEL) was used. The EC-SEBRIS faculty worked on this tool during the first year of the program to adapt the tool to match the targeted competencies and foci of the EC-SEBRIS certificate objectives. In this measure, students were asked to rate how confident they are in their ability to use promotion, prevention, and intervention levels of competencies to support young children’s social and emotional development. Students rate their abilities to develop nurturing and responsive relationships, create high quality environments, provide targeted social emotional supports, and implement intensive interventions. Each of these four general areas includes items relating to knowledge, skills, and reflective practice that promote social emotional competence in young children.

The survey has 60 items needed to be rated on a Likert scale of 1–5 (1 = Not at all confident; 2 = A little confident; 3 = Somewhat confident; 4 = Fairly confident; and 5 = Very confident.

The Nurturing and Responsive Relationships (NRR) domain is comprised of 24 items (1–24) focusing on the quality of the relationships formed with the child and their family (Tier 1). The following is a list sample items from this domain: “Develop meaningful relationships with children and families”; “Use a variety of strategies for building relationships with all families”; “Withhold judgment and feel empathy for parent/caregiver’s perspectives”; “Help parents/caregivers to appreciate the uniqueness of their young child”; “Show sensitivity to individual children’s needs and adapt and adjust accordingly (instruction, curriculum, materials, etc.)”; “Assess children’s strengths across all developmental and behavioral dimensions”; “Practice responsive caregiving based on the understanding each child’s unique development, responding to the child’s cues and signals, and following the child’s lead”; “Model appropriate expressions, labeling children’s emotions, and self-regulation”.

High Quality Environment (HQE) is the second domain in this survey (Tier 1). It consists of 19 items (25–43) addressing such issues as: “Asses children’s development and learning, including cognitive and language skills, social emotional development, approaches to learning, and health and physical development”; “Observe and document children’s work, play behaviors and interactions to assess progress”; “Manage behaviors and implement classroom rules and expectations in a manner that is consistent and predictable”; “Design activities to promote engagement (e.g., varies topics and activities in large and small groups from day to day, varies speech intonation to maintain children’s interest; modifies plans when children lose interest uses peers as models)”; “Provides children with varied opportunities to learn to understand, empathize with, and take into account other’s people perspectives”; “Teach staff to support children’s development of friendships and provide opportunities for children to play and learn from each other”.

Social Emotional Support (SES) is the third domain in the Self-Efficacy survey and addresses the promotion of prosocial behaviors and social emotional skills (Tier 1). There are (44–52) nine (9) items, which include the following: “Help children talk about their own and others’ emotions”; Provide children with varied opportunities to develop skills for entering into social groups, developing friendships, learning to help, and other pro-social behavior”; “Establishes and enforce clear rules, limits, and consequences for behavior”; “Teach staff to assist children in resolving conflicts by helping them identify feelings, describe problems, and try alternative solutions”.

The last domain in the survey is Intensive Interventions (II). This domain includes 8 items (53–60) addressing the ability to provide intervention to children who exhibit challenging behaviors (Tiers 2 and 3). For example, students were asked to rate their ability to “Team with a family to develop support plans for intensive interventions”; “Assess the presence and extent of atypical child behavior that may be a barrier to intervention and progress”; “Work as a team to develop and implement an individualized plan that supports inclusion and success with children with persistent, serious, challenging behavior”; “Assess problem behaviors in context to identify its function, and then devise interventions that are comprehensive in that they make the problem behavior irrelevant, inefficient, and ineffective”. The Self-Efficacy scales had very high internal reliability, with Cronbach’s α = 0.95, 0.96, 0.93, and 0.94 for NRR, HQE, SES, and II at baseline and Cronbach’s α = 0.93, 0.96, 0.93, and 0.92 at follow-up.

#### 2.2.4. Parent Survey

A Likert-scale survey was developed for parents, whose child received intervention, a targeted individualized behavior regulation plan, throughout their time in the EC-SEBRIS Certificate Program, to report the perceived change in their child. The survey consisted of items following the Desired Results Developmental Profile (DRDP) standards for social emotional development. Parents were asked to rate their child’s social competencies and behavioral challenges to determine whether their children were impacted from the students’ enrollment in the EC-SEBRIS Certificate Program, once in the beginning of the first semester and once at the end of the second semester (pre/post). The Likert 12-item scale has 4 levels of responses: 1 = Strongly disagree; 2 = Disagree; 3 = Agree; and 4 = Strongly agree. For example, parents were asked to indicate if they agree with the following: “Child understands that own physical characteristics and preferences are separate from others”; “Child shows awareness of other’s feelings and responds to other’s feelings appropriate”; “Child uses socially appropriate ways to stop self from acting impulsively”; “Child develops understanding of taking turns”; Child interacts cooperatively with other children through play”; Child forms friendships with other children”; “Child learns to understand the needs of other children and tries to follow social rules”. The parent survey had good internal consistency, with Cronbach’s α = 0.90 and 0.88 at baseline and follow-up, respectively.

### 2.3. Procedure

Human Subjects approval was granted following the review of the research protocol to the EC-SEBRIS certificate program. All participants (students, parents, and supervisors) were given a consent form to sign prior to their participation in the evaluation process.

Incoming students, during an orientation retreat in the summer, were informed about the program and its evaluation process and were given a consent form to sign in order to participate in the evaluation process of the program. The voluntary participation in this study was emphasized. Therefore, students were aware that the decision of whether or not to participate would not influence their future relations with or status in the certificate program. Certificate students were assured that their grades would not be affected by their decision to participate. In addition, students were informed that they are free to withdraw consent and to discontinue participation at any time without penalty or loss of benefits.

Confidentiality was maintained by removing personal information on the surveys. Instead numerical codes were given so pre- and post-scores could be matched without the need to identify the student. All surveys were kept in a locked cabinet in a locked room and were available only to the SDSU program evaluators.

Self-Efficacy survey was administered twice (pre–post) at the beginning of the program (August/September) and then at the end of the program (April/May). Supervisor surveys were self-administered to the supervisors at the beginning of fall semester and at the end of spring semester. Parent surveys were administered in the beginning of the school year and once at the end of the school year. Parents were given the surveys and completed them at home. They returned the survey in a sealed envelope.

Observations using the Arnett Caregiver Interaction Scale were administered by a trained evaluator. The evaluator/observer was trained to use the scale and was compared to faculty and other students trained to use it as well. Training was extensive and lasted until the inter-rater agreement reached 80%. Initial observations took place from September to November and the follow-up observation took place towards the end of the program (March–May). Observations took place at students’ work/volunteer sites while they interacted with the child/children for whom they were creating a regulation, prevention and/or intervention plan.

## 3. Results

In order to understand the results, it is important to clarify that the evaluation process started a year following the implementation of the program. Each cohort, as presented in Table 2 (Demographics by Cohort), was comprised of 22–35 students whose ethnicity, undergraduate major, and mental health experience contributed to the similarities and differences among the cohorts. For example, 48% of Cohort 2’s students were Hispanic, compared with only 22.9% in Cohort 1; 62.5% of the students in Cohort 4 had Child and Family as their undergraduate major, while around 48% of students from the other Cohorts earned their undergraduate degree majoring in Child and Family; 48% of students in Cohort 4 indicated having experience in mental health vs. only 18–21% of students in the other Cohorts. The evaluation of the EC-SEBRIS focused on the changes measured in the pre- and post-observations and survey reports; however, analyses were performed considering the possibility of differences between cohorts, which were independent groups. Analyses were performed with full factorial models, including the interaction of Cohort by Time (pre-post). However, with respect to the primary purposes of this paper, the main effect of Time was of principal interest.

### 3.1. Question 1

Did the students enrolled in EC-SEBRIS Certificate Program show an increase in their ability to provide sensitive, responsive care and support behavioral and emotional regulation in young children?

#### 3.1.1. Effects of Training on Students’ Competencies

Three sources of data were used in order to answer this question: direct pre-post observation of the students using the Arnett Scale, pre-post surveys of the site directors, and students’ pre-post self-efficacy report.

#### 3.1.2. Arnett Caregiver Interaction Scale (CIS)

A series of mixed-effects Analysis of Variance analysis, ANOVAs nesting students within cohorts were performed to assess the effects of Time and Cohort (1–4) on CIS scores. There were significant effects of Cohort on every scale. More importantly, results showed significant main effects of Time on Quality, Sensitivity, and Detachment. Additionally, there were significant interactions of Time by Cohort on Harshness and Permissiveness (see Table 2).

#### 3.1.3. Supervisor and Director Surveys on Student’s Performance

A mixed-effects hierarchical linear model was performed to assess the effects of Time and Cohort (1–4) on mean Supervisor Evaluations, where student was nested as a random factor within Cohort and supervisor was specified as a secondary random factor. There were 64 supervisors to 100 students with the average supervisor rating 2 students, and the maximum number of ratings from a single supervisor was 23. Models were performed with and without the interaction and compared using the Δ*χ*^2^ test to determine whether the interaction significantly improved the model. The Δ*χ*^2^(3) = 0.62, *p* = 0.892 demonstrated that the pithier model without the interaction was statistically superior; therefore, it was interpreted. The model was significant, Wald *χ*^2^(4, *N* = 139) = 25.42, *p* < 0.001. There was also a significant main effect of Time, *B* = 0.179, *z* = 3.46, *p* = 0.001, demonstrating that students received higher evaluations post-program.

Supervisors in their comments indicated that students improved in their knowledge and skills supporting children’s development and dealing with challenging behaviors: “The EC-SEBRIS student showed lots of growth during his time here. His biggest area of growth was building relationships with parents and caregivers”; “I feel that the student gained experience working directly with families, teachers and children. She excelled in building bonds with parents as her role was to meet with them, provide resources, and create goals”; “The student has a much stronger understanding of social emotional development and is a more competent parent educator, specifically for in-home behavioral support services”; and, “The student has grown exponential. She started the year out with infants and did such an amazing job. She really tuned into the children and their needs and has such a gentle and caring demeanor with them”.

#### 3.1.4. Self-Efficacy (SE)-Student’s Self Report

A series of mixed-effect ANOVAs nesting students within cohorts were performed to assess the effects of Time and Cohort (0–4) on mean SE domains. The effect of Time was significant on all subscales of the SE measure, as well as the Total (see Table 3). Although there were significant interactions of Cohort by Time for most scales and the Total (see Figure 2), these interactions were quantitative in nature, meaning some cohorts improved slightly more than others, although all cohorts improved.

The effect of Time was significant on all subscales of the SE measure, as well as Total; *F*(1, 270) = 415.42, *p* < 0.001, and Cohort, *F*(4, 270) = 9.71, *p* < 0.001, and a significant interaction, *F*(4, 270) = 3.73, *p* = 0.006 (see Figure 2). The effect of Time demonstrated that averaged across Cohort, SE Total increased from Pre- (M = 3.38) to Post-Program (M = 4.66) (see Table 3).

Although the Cohorts did differ from one another on the SE measures averaged across time, results indicate that students enrolled in the EC-SEBRIS certificate program in all cohorts have increased in their self-efficacy. Any interactions that existed were quantitative in nature, indicating that some Cohorts experienced greater increases in self-efficacy than others, but all increased. Students perceived themselves at the end of the program as better able to develop meaningful relationships with children and their families, to interact and respond sensitively to children meeting their needs, to create a safe environment where children’s needs are met, to partner and work with families respecting their cultural beliefs, to assess, observe, document, and design activities and interactions that can promote the child’s social emotional skills, to structure routines and provide environments where the child can engage, interact, communicate, learn, and feel competent, and to help children to socialize and interact with peers and other adults.

Overall, all Self Efficacy (SE) measures showed significant improvement across the duration of the program averaged across all cohorts.

### 3.2. Question 2

Did children improve in their social-emotional functioning after the intervention was offered to them by the students as reported by parents?

#### Parent Surveys on Child’s Social Emotional Skills

A mixed-effects ANOVA was performed to assess the effects of and Cohort (1–4) on mean Parent Survey scores. Results showed a significant main effect of Time, *F*(1, 113) = 10.03, *p* = 0.002, but no effect of Cohort, *F*(3, 113) = 0.98, *p* = 0.405, or the interaction, *F*(3, 113) = 0.91, *p* = 0.440. The effect of Time demonstrated that across all cohorts parents provided higher evaluations at Post-Program (M = 3.25) than Pre-Program (M = 3.01).

Parents recognized the improvement in their child’s social competencies and behaviors. Parents indicated that their child was able to regulate their emotions and behaviors, stopped biting, started using their words, had fewer tantrums, and were able to use social skills to befriend the children around them. Some examples of comments from parents are as follows: “My child has made a lot of progress from the pre-survey. For example, I have noticed that she shares toys with others and asks other children if she can borrow toys from them. Additionally, I am noticing that she is more familiar with emotions”; “I have noticed that he is able to calm himself down when he became frustrated or upset. He verbalizes his emotions and is able to name his feelings. He is aware that he has the ability to calm himself down, leading to more cooperative interactions”; “My child is literally a different person! She has become less violent and more compassionate towards other people. Prior to this program, my daughter had no friends and other kids were afraid of her. Now, she has friends and the kids actually want to play with her”.

## 4. Discussion

This paper describes a preventive approach to early childhood mental health with a focus on a professional development program (EC-SEBRIS) that prepares a workforce of early childhood professionals to provide relational sensitive care and to support emotion and behavior regulation in young children. Behavioral challenges occur often in classrooms, lead to high preschool expulsion rates, and can lead to long-term mental health problems. This primary prevention approach targets children and their families before an intervention is needed as it equips ECE professionals who work with young children on a daily basis with the skills and competencies to support children’s healthy development, address problem behaviors on-site, and provide them with a safe, familiar adult who is able to “contain” their frustrations and temperaments and help them learn effective coping skills to handle stress.

The EC-SEBRIS program addresses a vital need that early childhood professionals be trained to recognize and respond to early signs of social-emotional and mental health problems [2]. The program uses a wraparound triple method of teaching recommendations for infant family and early mental health services [73] that includes core content coursework, practicum experiences, and reflective supervision. Core content coursework is focused on attachment and affect regulation theories, assessment, and screening for typical and atypical development, as well as best practices of intervention strategies (e.g., Positive Behavior Support- PBS, Applied Behavior Analysis-ABA) that can help mitigate the effects of stress and address emotional and behavioral challenges in young children. Practicum experiences involved 20 h a week providing direct services/education to young children, and since most of the students had direct service jobs, their practicum classes allowed them to apply the knowledge and skills learned in their coursework to current behavioral problems and challenges they faced at their workplace. The EC-SEBRIS also followed recommendations that training of early childhood professionals include greater access to mental health consultation and reflective supervision [5,53,54,55,63] as the program included individual and group sessions with a licensed mental health professional, and use of videotaping and on-site coaching to further students’ abilities to address social-emotional and behavioral challenges in young children.

This study provides support that the EC-SEBRIS program was successful in promoting positive outcomes for the ECE professionals (students) enrolled in the program and for the children and families they served. Observations of students interacting with young children at their practicum sites indicate that the students increased in Sensitivity scores on the Arnett Caregiver Interaction Scale. That is, they appeared to have improved in their positive interactions with children: speaking more warmly to them, listening attentively, and enjoying being with them. The results provide evidence that the students increased in behaviors that encouraged children to explore, learn new things, and exhibit prosocial behaviors.

Further evidence that the students increased in their abilities to provide behavior support and enhance social emotional development in young children is provided by surveys given to their site supervisor. Analyses of scores on the Goal Achievement Scale for supervisors revealed that students improved on teaching competencies that include a greater understanding of children’s social and emotional development and improvement on responding appropriately to children who are distressed. Comments of the supervisors indicate that students improved in their ability to support challenging behaviors in the classroom, connect with parents and families, and provide children with positive and sensitive care: “She (student) really tuned into the children and their needs and has such a gentle and caring demeanor with them”.

Other evidence of positive changes in the students is provided by their self-report of their confidence in demonstrating competencies needed to promote social and emotional development and to address challenging behaviors in young children (National Center on the Social and Emotional Foundations for Early Learning—CSEFEL). Results examining self-efficacy scores suggest students’ knowledge and confidence improved in developing nurturing and responsive relationships, creating high quality environments, providing targeted social emotional supports, and implementing intensive interventions. That is, at the end of the program, students perceived themselves as more capable in competencies such as developing meaningful relationships with children and their families, structuring routines and providing environments that help young children learn, feel competent, and socialize with peers and adults, partnering and working with families from diverse backgrounds, and implementing assessments, observations, and intensive intervention activities.

In addition to providing support that the EC-SEBRIS program was successful in promoting positive outcomes for the ECE professionals (students) enrolled in the program, this study provides evidence that the program helped the children and families that the ECE professionals (students) served. Similarly, numerous studies on positive behavior support in ECE settings indicate that effective behavior management in the classrooms support young children’s abilities to regulate their emotions and behaviors and yield better child outcomes [47,48,49,50]. The Parent surveys assessing their children’s social competencies and behavioral challenges described by Desired Results Developmental Profile (DRDP) standards show that parents perceived positive social emotional gains for their children. Analyses revealed that parents recognized the improvement in their child’s social competencies and behaviors, as they indicated that their child is able to regulate their emotions and behaviors, stopped biting, have less tantrums, and is able to use social skills to befriend the children around them. In open-ended comments, parents indicated the change in their child’s social skills in the wake of the intervention. Parents wrote that their child is able to share, identify, and express emotions, problem-solve, manage strong emotions, and communicate needs. For example, according to one parent: “My child is literally a different person! She has become less violent and more compassionate towards other people. Prior to this program, my daughter had no friends. Now, she has friends and the kids actually want to play with her”. Hence, positive sensitive teacher–child relationships appear to decrease behavioral challenges and the negative effects for children at risk for externalizing and internalizing problems [38,46].

The results presented on the EC-SEBRIS certificate program show that intensive training that combines knowledge, experience, and reflective practice support the professional growth in early childhood professionals and their ability to provide nurturing, sensitive, and responsive care to children. Analyses of behavioral observations of the ECE professionals (students) interacting with young children and site supervisors’ report of the ECE professionals’ competencies offer evidence that the program had positive effects on the ECE professionals’ abilities to support young children’s socio-emotional and behavior regulation. ECE professionals also gained a sense of self efficacy in their ability to understand behaviors and provide children with the appropriate support and regulation plan. In addition, the results suggest that the positive effects extended to the children served by the ECE professionals in the EC-SEBRIS program. That is, analyses of parent reports of their children’s behaviors, as well as their open-ended comments, provide evidence that the children served by the ECE professionals made positive gains over the one-year EC-SEBRIS program.

Future research is needed to further examine the effects of the EC-SEBRIS program on the children served. Assessments that measure children’s behaviors with targeted periodic observations in the classroom as well as at home are needed to confirm the current findings that were based solely on parent report. In addition, most of the participants in the EC-SEBRIS program, as well as in the field of early childhood education in general, are female. Future research should examine the generalizability of the current results and include a larger sample so that we can understand the characteristics of participants (e.g., gender, years of work in early childhood education, experience with mental health professionals) and children served (e.g., gender, age), which may influence the effectiveness of the EC-SEBRIS program. Additionally, future studies should include a control group, and the long-term effects of the training on the ECE professionals and the children they served should be examined.

## 5. Conclusions

The EC-SEBRIS program presented above was designed to prepare a workforce of early childhood professionals to provide sensitive relational care and to address early mental health problems with behavior and emotional regulation. The one-year graduate certificate training program includes core content coursework, practicum experiences, and reflective supervision that was provided in individual and group sessions by an experienced licensed mental health professional. The use of videotaping for supervisors to provide instruction and encourage reflection, as well as on-site coaching at their practicum sites, were components facilitating the integration of theory to practice and skill building in the ECE professionals.

The EC-SEBRIS wraparound training and coaching model views the early childhood teacher/care educator as an important member of the first-response team to the child’s needs and daily care. The EC-SEBRIS training equipped the teacher with the knowledge and skills to recognize the child’s needs and come up with a response, which meets the child’s needs. It is built on strength-based framework [81] in which teachers are trained to identify social competencies and capitalize on the strengths of children with challenging behaviors early on, rather than focusing on their problem behaviors. The program also recognizes that the stress, burnout, and frustration early childhood educators feel in their jobs affect their relationships with the children in their care and the environments they create for the children [51,52]. Thus, the program enhances ECE professionals’ capacity to provide warm, responsive caregiving and reduce problem behaviors and rates of expulsion by giving them greater access to mental health consultation [5,53,54,55], as well as giving them direct training to increase their knowledge, skills, and competencies to support emotion and behavior regulation in young children under their care. The program equips the ECE professionals who work with young children on a daily basis with the skills and competencies to support children’s healthy development, to address problem behaviors on-site, and to provide children with a safe, familiar adult who is able to “contain” their frustrations and temperaments. It also provides other teachers on-site with an opportunity to learn from their peers and model their behaviors and problem-solving practices. This study renders support to the assertion that professional training that focuses on emotional and behavior regulation support is essential for early childhood professionals working with young children [68].

## Figures and Tables

**Figure 1 brainsci-07-00120-f001:**
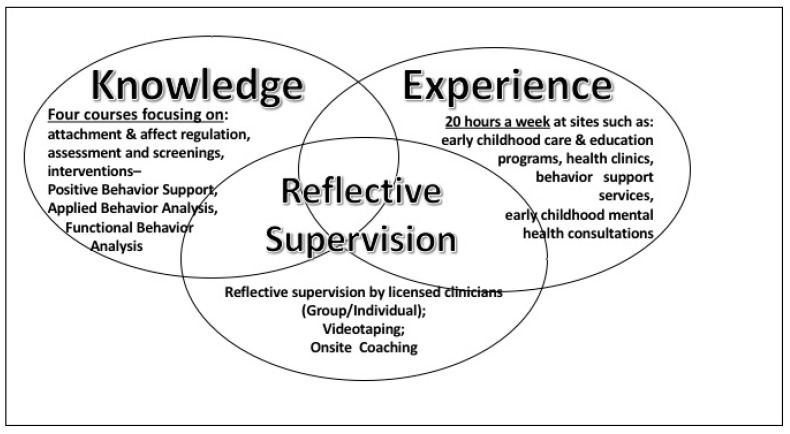
Early Childhood Social Emotional and Behavior Regulation Intervention Specialist (EC-SEBRIS) Certificate Triple Mode Teaching Model.

**Figure 2 brainsci-07-00120-f002:**
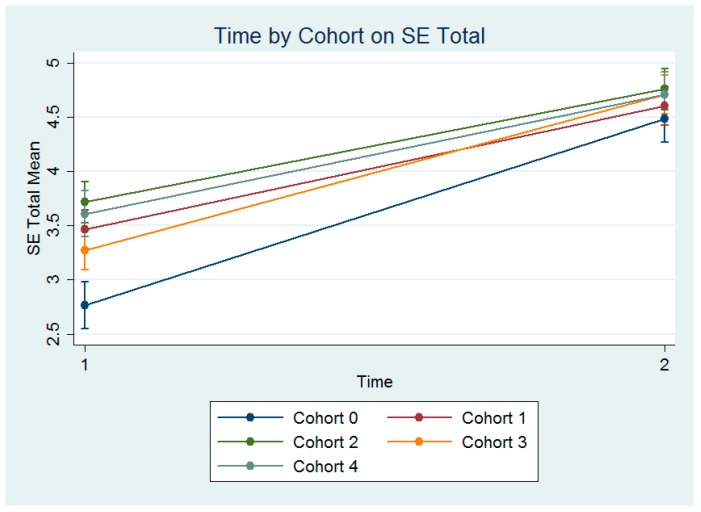
The Effects of Time and Cohort on Self Efficacy (SE) Total Scores.

**Table 1 brainsci-07-00120-t001:** Participant Demographics.

Item	Cohort 0		Cohort 1		Cohort 2		Cohort 3		Cohort 4	
	Valid %	N	Valid %	N	Valid %	N	Valid %	N	Valid %	N
**Ethnicity**										
White	44.44	12	48.6	17	32.0	8	57.6	19	41.7	10
Hispanic	40.74	11	22.9	8	48.0	12	30.3	10	37.5	9
Black	7.41	2	11.4	4	-	-	-	-	4.2	1
Other	7.41	2	17.2	6	20.0	5	12.1	4	16.7	4
**Undergraduate Major**										
Child and Family	40.74	11	48.3	17	48.0	12	48.5	16	62.5	15
Social Science (Other)	14.81	4	31.4	11	36.0	9	33.3	11	25.0	6
Language	14.81	4	5.7	2	-	-	9.1	3	-	-
Communication or Arts	11.11	3	-	-	-	-	6.1	2	4.2	1
Other	18.52	5	14.3	5	16.0	4	3.0	1	8.3	2
**Mental Health Experience**										
Yes	18.52	5	18.2	6	20.0	5	21.2	7	45.8	11
No	81.48	22	81.8	27	80.0	20	78.8	26	54.2	13
Bilingual										
Yes	18.52	5	37.1	13	60.0	15	33.3	11	27.5	9
No	81.48	22	62.9	22	40.0	10	66.7	22	62.5	15
**Primary Language**										
English	85.19	23	62.9	22	56.0	14	75.8	25	87.5	21
Spanish	14.81	4	20.0	7	32.0	8	24.2	8	8.3	2
Other	-	-	17.1	6	12.0	3	-	-	4.2	1
**Second Language**										
English	-	-	100	13	66.7	10	72.7	8	33.3	3
Spanish	80.0	4	-	-	13.3	2	18.2	2	55.6	5
Other	20.0	1	-	-	20.0	3	9.1	1	11.1	1

**Table 2 brainsci-07-00120-t002:** The Effects of Time, Cohort, and Time × Cohort on Arnett Caregiver Interaction Scale (CIS).

	F (dfd = 113)	*p*	R^2^	M Pre/Post
**Quality**				
Time (dfn = 1)	67.12	<0.001	0.373	
Cohort (dfn = 3)	6.7.9	<0.001	0.153	3.15/3.35
Time × Cohort (dfn = 3)	0.35	0.786	0.009	
**Sensitivity**				
Time (dfn = 1)	98.57	<0.001	0.466	
Cohort (dfn = 3)	75.61	<0.001	0.667	2.59/2.95
Time × Cohort (dfn = 3)	0.25	0.863	0.023	
**Harshness**				
Time (dfn = 1)	2.34	0.128	0.005	
Cohort (dfn = 3)	45.68	<0.001	0.474	1.25/1.20
Time × Cohort (dfn = 3)	2.97	0.035	0.023	
**Detachment**				
Time (dfn = 1)	30.81	<0.001	0.214	
Cohort (dfn = 3)	9.49	<0.001	0.201	3.51/3.76
Time × Cohort (dfn = 3)	0.41	0.744	0.011	
**Permissiveness**				
Time (dfn = 1)	1.48	0.226	0.013	
Cohort (dfn = 3)	151.64	<0.001	0.801	2.97/3.01
Time × Cohort (dfn = 3)	9.75	<0.001	0.206	

dfd = degree of freedom in the denominator; dfn = degree of freedom in the nominator.

**Table 3 brainsci-07-00120-t003:** The Effects of Time, Cohort, and Time × Cohort on Self-Efficacy.

	F (dfd = 135)	*p*	R^2^	M Pre/Post
**SE NRR**				
Time (dfn = 1)	413.11	<0.001	0.754	3.55/4.67
Cohort (dfn = 4)	9.10	<0.001	0.212	
Time × Cohort (dfn = 4)	5.13	<0.001	0.132	
**SE HQE**				
Time (dfn = 1)	398.63	<0.001	0.747	3.41/4.67
Cohort (dfn = 4)	6.84	<0.001	0.169	
Time × Cohort (dfn = 4)	4.54	0.002	0.119	
**SE SES**				
Time (dfn = 1)	342.68	<0.001	0.717	3.46/4.72
Cohort (dfn = 4)	6.02	<0.001	0.151	
Time × Cohort (dfn = 4)	3.81	0.006	0.101	
**SE II**				
Time	465.97	<0.001	0.775	2.75/4.53
Cohort	3.52	0.009	0.094	
Time × Cohort	1.58	0.18	0.045	
**SE Total**				
Time (dfn = 1)	493.71	<0.001	0.785	3.38/4.66
Cohort (dfn = 4)	8.38	<0.001	0.199	
Time × Cohort (dfn = 4)	4.44	0.002	0.116	

Note: dfd = degree of freedom in the denominator; dfn = degree of freedom in the nominator; SE = Self-Efficacy; NRR = Nurturing and Responsive Relationships; HQE = High Quality Environment; SES = Social Emotional Support; II = Intensive Interventions.

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
