# Peer review of "Investing in the Early Childhood Mental Health Workforce Development: Enhancing Professionals’ Competencies to Support Emotion and Behavior Regulation in Young Children"

_brainsci, 2017, doi:10.3390/brainsci7090120_

Round 1

Reviewer 1 Report

The reader would benefit if the following paragraph appeared earlier in the paper (p. 2): "The purpose of this paper is . . ."

Throughout the paper, different terms are used to describe the same individuals: students, trainees, participants. Please decide on only one term and use it consistently.

The Venn diagram does not have any overlapping points and, therefore, a Venn diagram is not suitable for graphically presenting the model.

Page 5: line 230. It appears if there were -35 students who graduated.

It is not clear who completed the Caregiver Interaction Scale (CIS).

Page 7, line 308. Is "Developing meaningful relationships with children and families" the name of the domain or the first of the items? The reason this is not clear is because all the other sample items start with the imperative (withhold, help, show, assess, etc.).

Page 3, lines 102 to 106. Please edit for clarity.

The term "Cohort by Time" is not explained and it is an important term for understanding the tables. Does "time" refer to the pre-test and the post-test? Is "time" the length of the study or the practicum?

The table presentation is in need of revision. First, while Time, Cohort, and their interaction are described in the preceding paragraphs, repeating that information in a caption adjacent to each table will ease readers’ interpretation of these tables. The presentation of the means of the subgroups seems to indicate that Cohort test produced means for a pre- and post-test rather than the Time category. Ultimately, the table presentation may be made more clear by separating both of them into 3 tables; one table for Time, for Cohort, and for their interaction in each of the 5 categories. Then the means and metrics of each group being evaluated can be more accurately displayed.

Author Response

Please see the attached PDF for the responses to your great comments.

Reviewer 2 Report

The article reads well and is interesting with specific examples of how the intervention helped the staff to improve their work with children.

I was a bit confused with the language of teacher, educator and professional so think this could be made clearer and terms used consistently.

I also wondered whether there is any data that could be included in relation to the children's behaviour - other than self-reports from parents and staff. For example did the early childhood service keep any data related to behavioral concerns or referrals for support in relation to challenging behaviours? 

Author Response

Thank you for your constructive comments. 

We revised the article to be more concise and consistent with its use of language to describe the target population.

We have data on the children based on parents and staff. We addressed it in the limitations that were added to the discussion.

We added a conclusion section at the end with key points.